# Pituicytoma Associated with Suspected Cushing’s Disease: Two Case Reports and a Literature Review

**DOI:** 10.3390/jcm11164805

**Published:** 2022-08-17

**Authors:** Tongxin Xiao, Lian Duan, Shi Chen, Lin Lu, Yong Yao, Xinxin Mao, Huijuan Zhu, Hui Pan

**Affiliations:** 1Key Laboratory of Endocrinology of National Health Commission, Department of Endocrinology, State Key Laboratory of Complex Severe and Rare Diseases, Peking Union Medical College Hospital, Chinese Academy of Medical Science and Peking Union Medical College, Beijing 100730, China; 2Eight-Year Program of Clinical Medicine, Peking Union Medical College Hospital, Peking Union Medical College, Chinese Academy of Medical Sciences, Beijing 100730, China; 3Department of Neurosurgery, Peking Union Medical College Hospital, Chinese Academy of Medical Sciences and Peking Union Medical College, Beijing 100730, China; 4Department of Pathology, Peking Union Medical College Hospital, Chinese Academy of Medical Sciences and Peking Union Medical College, Beijing 100730, China

**Keywords:** pituicytoma, Cushing’s disease, pituitary adenoma, diabetes insipidus

## Abstract

(1) Background: Pituicytomas are rare gliomas located in the neurohypophysis or infundibulum. A misdiagnosis of pituicytoma as pituitary adenoma is common because of similar location and occasional endocrine disturbances. (2) Case presentation: We present two cases with the comorbidity of pituicytoma and Cushing’s disease (CD). Case 1 is that of a 51-year-old woman, the first reported case of the comorbidity of pituicytoma, CD, and central diabetes insipidus. She received a diagnosis of CD and central diabetes insipidus. After transsphenoidal surgery, histopathology confirmed the diagnosis of pituicytoma and adrenocorticotropin-secreting microadenoma; case 2 is that of a 29-year-old man who received a biochemical diagnosis of CD, but he received a histopathological confirmation of only pituicytoma. Both patients achieved a remission of hypercortisolism without relapse during the follow-up, but they developed hypopituitarism after surgery. We also reviewed all published 18 cases with the comorbidity of pituicytoma and any pituitary adenoma. (3) Conclusions: Pituicytoma might present pituitary hyperfunction disorders such as CD or acromegaly, with or without pathologically confirmed pituitary adenoma. CD is the most common hyperpituitarism occurring concurrently with pituicytomas. The remission rate and hypopituitarism after surgery seem similar or slightly lower in CD than in common pituitary adenomas, but the long-term prognosis is unexplored.

## 1. Introduction

Pituicytoma is a rare glioma (World Health Organization (WHO) grade I) that originates in the posterior pituitary gland or pituitary stalk [1], with <200 cases reported to date [2]. A misdiagnosis of pituicytoma as pituitary adenoma or craniopharyngioma is common before surgery because pituicytomas are frequently located in the sellar or suprasellar region (while they can develop anywhere along the hypothalamic–posterior pituitary axis [2]), and their clinical presentations, such as visual defect and headache, are also nonspecific. Furthermore, some pituicytoma patients with negative immunopathologic results in any pituitary hormone staining test [1] show evident endocrine disturbances, which mimic pituitary adenoma in clinical presentation. Some endocrine alterations, such as anterior pituitary hormone deficit, mild hyperprolactinemia, and diabetes insipidus, can be explained by the mass effect or stalk effect of posterior pituitary tumors (PPTs) [2,3]. However, remarkable pituitary hyperfunction disorders, mostly Cushing’s disease (CD), have also been reported in 17 patients [4,5,6,7,8,9,10,11,12,13,14,15,16], most of whom had a suspected diagnosis of pituitary adenoma before surgery. In patients presenting with biochemical CD, concurrent adrenocorticotropin (ACTH)-secreting adenoma or hyperplasia was occasionally evidenced in histopathological examinations [4,7,9,14,15], but mostly, only pituicytoma was identified. Whether pituicytoma played a role in the increase in specific pituitary hormones remains unknown.

Here, we report two cases of comorbid pituicytoma and biochemically diagnosed CD. In case 1, ACTH-secreting pituitary adenoma and pituicytoma comorbidity was histopathologically diagnosed, whereas in case 2, only pituicytoma was histopathologically diagnosed. Furthermore, in case 1, central diabetes insipidus was also diagnosed; thus, it is the first reported case of pituicytoma, CD, and diabetes insipidus comorbidity to our knowledge. Moreover, previous case reports were reviewed in this study to summarize the clinical presentations, radiological features, pathologic features, and prognosis. We hope our study highlights pituicytoma as a differential diagnosis of sellar mass, even when the clinical presentation is typical of CD.

## 2. Case Presentation

We reviewed the records of patients who underwent pituitary surgery and received a diagnosis of pituicytoma after pathological examination at our hospital between 2013 and 2020. Among the six patients who received a pathological diagnosis of pituicytoma, two patients received a biochemical diagnosis of CD before surgery. Table 1 and Table 2 present the details of pituitary hormone levels in the two patients before and after surgery, respectively.

Case 1: A 51-year-old woman had headache, hypertension, nausea, palpitation, and hirsutism for 15 years, and she presented with increasing asthenia, nausea, insomnia, ostealgia, and diabetes insipidus for 6 months. First, she visited another hospital. Her blood pressure was approximately 150–170/80–100 mmHg when on several antihypertensive drugs. Her midnight serum cortisol was 303.1 nmol/L (11.0 µg/dL), and her ACTH level was 10.02 pg/mL. Her overnight 1 mg dexamethasone suppression test showed no suppression of serum cortisol (4.0 µg/dL). Hence, she received a diagnosis of Cushing’s syndrome. The magnetic resonance imaging (MRI) of the pituitary gland showed no evident lesion except for a thickened pituitary stalk (5.5 × 4.7 mm). Adrenal computed tomography showed a nodule in the left adrenal gland. Then, she underwent left laparoscopic adrenalectomy in that hospital, and adrenocortical hyperplasia was identified through histological examinations. Due to the persistence of diabetes insipidus and increased ACTH, she was referred to our department 2 months after the adrenalectomy.

Her physical examination revealed moon face, central obesity, abdominal striae, and lower extremity edema. Her height was 164 cm, and her weight was 83 kg, with a body mass index (BMI) of 30.86 kg/m^2^. She had a history of hypertension, impaired glucose tolerance, and hyperlipidemia and had 5 months of menopause. Her blood pressure was approximately 130/80 mmHg when on a daily dose of 30 mg of nifedipine. Her midnight serum cortisol was 17.37 µg/dL, and her ACTH level was 55.7 pg/mL (reference range: 0–46 pg/mL), with 24 h urinary-free cortisol (24 h UFC) of 186.53 µg (reference range: 12.3–103.5 µg). An overnight 1 mg dexamethasone suppression test showed unsuppressed serum cortisol (3.09 µg/dL), whereas a high-dose dexamethasone suppression test showed suppressed serum cortisol (1.09 µg/dL). Dynamic pituitary MRI revealed a 9.7 × 4.0 mm lesion at the left side of the pituitary gland, with a thickened pituitary stalk (4.3 × 2.9 mm) and the loss of T1 signal hyperintensity in the posterior pituitary gland (Figure 1). She received a diagnosis of CD based on these data. Furthermore, central diabetes insipidus was diagnosed based on a water restriction test: Urine osmolarity was 105 mOsm/kgH_2_O, serum osmolarity was 323 mOsm/kgH_2_O, and serum sodium was 156 mmol/L at 8 h water restriction. Urine output dropped from 4370 mL/24 h to 1440 mL/24 h after treatment with 0.1 mg desmopressin. However, the cause of diabetes insipidus and thickened pituitary stalk remained unknown. In transsphenoidal surgery, a soft white mass at the left side of the pituitary was resected, and a white-gray tough lesion in the posterior pituitary was sampled for biopsy. At 17 months of follow-up, she had no Cushingoid symptoms or sellar mass recurrence, with a thinner pituitary stalk (2.9 × 2 mm). As only a partial remission of diabetes insipidus was achieved, and hypopituitarism had developed, she was discharged with a prescription of 0.05 mg desmopressin twice a day, 1.25 mg prednisone daily, and 100 µg levothyroxine every day.

A histopathological examination of the 9 × 5 × 4 mm resected specimen (Figure 2) showed two different lesions, indicating adenoma and pituicytoma. The lesion diagnosed as pituitary adenoma was positive for T-box family member TBX19 (T-PIT) and ACTH, and its Ki-67 proliferation index was approximately 10%. The posterior pituitary lesion consisted of mostly spindle cells that were positive for thyroid transcription factor-1 (TTF-1), glial fibrillary protein (GFAP), and epithelial membrane antigen (EMA), which met the diagnosis criteria of pituicytoma. The staining of cytokeratin AE1/AE3 was positive, and diffuse cells expressed CD3, CD138, and CD38. The staining of any pituitary hormone was negative. Based on these results, the diagnoses of ACTH-secreting adenoma and pituicytoma were confirmed.

Case 2: A 29-year-old man with a 3-year history of hypertension and a 1-year history of central obesity, decreased libido, asthenia, and ostealgia was referred to our department. He had a history of renal calculus, rib fractures, and lumbar compressional fractures. An overnight 1 mg dexamethasone suppression test in another center could not suppress serum cortisol (16.6 µg/dL), and then he was referred to our center. His physical examination revealed moon face, central obesity, abdominal striae, and lower extremity edema. His height was 163 cm, and his weight was 61 kg, with a BMI of 22.96 kg/m^2^. His blood pressure was 170/125 mmHg. His midnight serum cortisol (21.25 µg/dL) and morning serum cortisol (19.7 µg/dL) showed impaired cortisol rhythmicity, and his ACTH was 37.5 pg/mL. Furthermore, his 24 h UFC was 185.8 µg. A low-dose dexamethasone suppression test could not suppress 24 h UFC (43.7 µg), whereas a high-dose dexamethasone suppression test suppressed 24 h UFC (20.8 µg). Dynamic pituitary MRI suspected a 3 × 4 mm hypointense nodule at the left side of the pituitary (Figure 3). To verify this finding, bilateral inferior petrosal sinus blood was sampled under a desmopressin acetate activation test. It confirmed an asymmetrical secretion of ACTH in the left part of the sellar region, as the central (left)/peripheral plasma corticotropin gradient without activation was 3.9 (66.8/17.2 pg/mL), and this gradient exceeded 55 (>1250/22.6 pg/mL) after 3 min of desmopressin activation. Hence, CD was suspected. The patient underwent endoscopic transsphenoidal surgery. A white-gray, soft, and poorly vascular lesion was noticed, and a gross total section was achieved. After surgery, the remission of hypercortisolism was achieved, with no relapse in 8 months of follow-up. The patient developed hypopituitarism, and hence, he was discharged with a prescription of 20 mg hydrocortisone acetate daily (in three divided doses), 25 µg levothyroxine daily, and 0.25 g testosterone undecanoate injection once a month.

The histopathological examination of the 5 × 4 × 3 mm white-gray tissue revealed TTF-1, S-100, and synaptophysin expression in the tumor, indicating the lesions to be pituicytoma (Figure 4). Some focal cells were positive for GFAP. Somatostatin receptor 2 (SSTR-2) and AE1/AE3 were negative in the staining analysis. Mitotic activity was absent. ACTH was positive for a few adjacent normal pituitary gland cells, but no corticotroph hyperplasia or adenoma was confirmed in the specimen. The diagnosis of CD was not finally confirmed in the histopathological study.

## 3. Review of the Literature

We searched PubMed and Embase from database inception to November 2021 for relevant studies by using the terms “pituicytoma” combined with “pituitary adenoma,” “Cushing,” “hypercortisolism,” or “acromegaly” (Figure 5). Only publications with the full text in English were reviewed. We included all cases of patients with confirmed comorbidity of pituicytoma and pituitary adenoma. Furthermore, cases of patients with sufficient data supporting a biochemical diagnosis of CD or acromegaly were included even when a histopathologic examination confirming pituitary adenoma was lacking. Cases of patients with neither histopathologically confirmed pituitary adenoma nor a clinically confirmed biochemical diagnosis of CD or acromegaly were excluded. In total, 18 patients were considered in this study, comprising 14 patients with the comorbidity of pituicytoma and confirmed CD, 3 patients with the comorbidity of pituicytoma and acromegaly, and 1 patient with the comorbidity of pituicytoma and nonfunctional pituitary adenoma. Table 3 presents the clinical and pathological characteristics of patients with CD included in this study.

### Overview: Pituicytoma Associated with Suspected Cushing’s Disease

Regarding patients with biochemically diagnosed CD, 14 cases (females: 11/14, 78.6%) have been reported in previous studies. Age at diagnosis ranged from 7 to 57 (mean: 41) years, with 11 of 14 patients aged >30 years. The mean follow-up time was 18 months, ranging from 0 to 96 months.

The clinical presentations of all patients consisted of an increased level of cortisol. Only one patient was reported to have a visual impairment (case 11). Diabetes insipidus was not reported in any of the patients with pituicytoma associated with pituitary adenoma.

In 92.8% (13/14) of the patients, sellar masses could be identified through MRI before surgery. Most lesions were <10 mm at the greatest dimension (mean: 7 mm), except for one patient (case 4, diagnosed with pituitary hyperplasia and pituicytoma). Pituitary stalk abnormality was reported in five patients with CD: two patients had a thick pituitary stalk, one patient had a short pituitary stalk, one patient had a slight swelling of the pituitary stalk, and one patient had a slight bulge on the pituitary stalk.

All patients underwent transsphenoidal surgery, and pituicytomas were identified histopathologically. Eight patients received a gross total resection, and two received a subtotal resection in their first surgery. Pituitary adenomas were confirmed in 5 patients (3 of 14 with CD). Of the 14 (64%) patients with CD, 9 patients achieved remission and no relapse without further reoperation or radiotherapy at an average 18-month follow-up. Among the three patients with confirmed ACTH-secreting adenomas, one received bilateral laparoscopic adrenalectomy, which revealed diffuse secondary adrenocortical hyperplasia, whereas two patients achieved remission without further intervention. Three of nine (33%) patients developed total or partial hypopituitarism, which required hormone replacement treatment in the long term.

Immunohistochemically, all the patients were positive for TTF-1 or S-100 and negative for hypophyseal hormones as per the diagnosis criteria of pituicytoma. Regarding other variable markers, pituicytoma cells were positive for GFAP in 7 of the 11 patients (63.6%) with CD. EMA was positive in three of the nine (33.3%) patients with CD. BCL-2 was positive in three of the four patients with a comorbidity of pituicytomas and CD, and synaptophysin was observed in three of the eight patients with CD. All the patients with CD had a Ki-67 proliferation index of pituicytoma ranging from <1% to 2%.

By far, studies on the comorbidity of pituicytoma and hyperpituitarism have mainly concentrated on clinical presentation and short-term prognosis. The knowledge of presurgical diagnosis, complications, long-term prognosis, and genetic background of pituicytomas associated with CD is very limited.

## 4. Discussion

### 4.1. Symptoms and Diagnosis: Pituicytoma and Pituicytoma Associated with Cushing’s Disease

To date, distinguishing pituicytoma from pituitary adenoma before surgery is difficult because of similar clinical presentations and nonspecific radiological features. Anterior pituitary deficiency (partial or total) and mild hyperprolactinemia, which are likely secondary to mass effect and pituitary stalk effect, respectively, are the most commonly observed in patients with pituicytoma [2,17]. Meanwhile, although pituicytoma cannot secrete hypophyseal hormone by itself, endocrinal disturbances have been reported in many cases, and a study stated it to be the second most common clinical symptom [2]. To date, a biochemical diagnosis of CD is the most common hyperpituitarism (14 cases) associated with pituicytomas, and acromegaly is the second (3 cases), irrespective of whether a functioning case of pituitary adenoma was histopathologically identified.

Preoperative diabetes insipidus was reported in approximately 5% of the patients with pituicytoma [2], and our case 1 was the first reported case of pituicytoma with concomitant CD and diabetes insipidus. Concomitant CD and diabetes insipidus are extremely rare. In our case, CD achieved remission after the surgical removal of ACTH-secreting microadenoma, but central diabetes insipidus did not achieve remission. Pituitary stalk involvement in pituicytoma possibly led to this dysfunction.

In terms of radiologic examinations, the size of pituicytoma associated with CD (mean: 7 mm, ranging from not clearly visible to 15 mm) seems much smaller than the major diameter of pituicytomas (mean: 22.99 mm, median: 20 mm, ranging from 4 to 70 mm) reported by the latest review [2]. The small lesions might also explain why visual impairment was rare in pituicytoma patients suspected with CD, although it is considered the most common symptom of pituicytoma [2]. Similarly, for common CD cases, approximately 80%–90% of lesions are also considered microadenomas (<1 cm) [18]. Since the imaging features of pituicytoma and common CD greatly overlap, it is difficult to make a differential diagnosis based on MRI.

The histopathological examination of pituicytoma mainly revealed spindle cells with abundant eosinophilic cytoplasm arranged in fascicles or storiform patterns. Pathological diagnosis is crucial for pituicytoma, and the criteria include positive staining for TTF-1, S-100, and vimentin and negative staining for any hypophyseal hormones and neuronal or neuroendocrine markers [2]. The expressions of GFAP, EMA, and synaptophysin have been variable, but whether they suggest variable clinical features is unclear. Notably, the unequivocal presence of ACTH-secreting pituitary adenoma was only confirmed in 3 of the 14 patients with suspected CD [7,14,15]. For the remaining cases, the diagnosis of CD was not finally confirmed histopathologically, consistent with case #2 in our report.

### 4.2. Treatment and Prognosis: Comparison with Cases with Typical Pituitary Adenomas

Surgical resection has been the gold standard for pituicytoma treatment by far. In patients with pituicytoma associated with CD, approximately 64% of the patients with pituicytoma achieved remission and no relapse of CD after the first surgery, irrespective of the presence of ACTH-secreting adenoma. Moreover, 93% of the patients could achieve remission after additional reoperation or radiotherapy. No recurrence or malignant transformation has been reported in the relatively short follow-up period (average of 18 months) [2]. In all the patients with common CD, the mean remission rate after the first removal of microadenoma (82.1%) was slightly higher than the patients with CD and pituicytoma, with a mean recurrence rate of 11.7% [18].

Substantial bleeding during pituicytoma surgery is common (up to 86.2% in a study) [2]. However, most cases with CD associated with pituicytoma did not clearly describe the vascularization of lesions or intraoperative bleeding, except for one case [9]. In case 1 of our study, the lesion was also described as a well-vascularized one, with approximately 200 mL intraoperative bleeding. However, in case 2, the lesion was a poorly vascularized one. Although intraoperative bleeding has not been reviewed in common CD, bleeding or hematomas are reported in up to 6% during the perisurgical period [18]. In brief, pituicytoma, including those associated with CD, seems to bleed easily during surgery, but a standardized description in reported cases is absent.

Anterior hypopituitarism is the most common complication after surgery in cases of CD associated with pituicytoma (3/9 in cases clearly describing complications). The two patients reported in our study also developed hypopituitarism. By comparison, after transsphenoidal surgery for common CD, 2%–40% may develop this insufficiency [18]. In brief, no significant difference in the hypopituitarism complication rate was noted between patients with or without pituicytoma. The risk of hypopituitarism and the long-term prognosis need to be studied.

In case 1, it is clear that she wrongly underwent an adrenalectomy procedure at first because of a diagnostic error mistaking the nodules in the adrenal gland as the responsible lesion. After surgery, the thickened pituitary stalk, which was possibly caused by pituicytoma infiltration, spontaneously became thinner (from 4.3 × 2.9 mm to 2.9 × 2 mm) without adjuvant radiotherapy. Direct damage during surgery might not significantly contribute to the remission of pituitary stalk thickening because only an adenoma resection and a posterior gland biopsy (but not a pituitary stalk biopsy) were performed. We presumed that the decreased blood supply in the pituitary stalk region after surgery is a possible cause. In this situation, the relatively ample vascularization, a frequently noticed feature of pituicytomas [19], which was also described in case 1, might rebuild later and help the neoplasm prominently recur during follow-up. Another assumption was hypophysitis, which might be secondary to sellar tumors and contribute to pituitary stalk thickening, reaching partial remission after surgery. Hypophysitis could arise secondary to sellar or para-sellar tumors, including germinomas, astrocytomas, craniopharyngiomas, and meningiomas [20]. Although no confirmed hypophysitis secondary to pituicytoma has previously been reported, it is a potential mechanism. Additionally, the resection of pituitary adenoma might somewhat inhibit hyperplasia or the proliferation of cells in the stalk area by disturbing the interaction between adjacent adenoma and pituicytoma. The close monitoring of the change in the stalk region in case 1 is needed.

### 4.3. Pituicytoma and CD: Possible Mechanisms

Regarding the possible mechanisms of the comorbidity of pituicytoma and pituitary adenoma, it should first be noted that there is a possibility they might have no underlying pathophysiological connection. Since pituitary adenomas are not rare, surgery in the pituitary region might lead to occasional findings of pituicytoma because of symptoms caused by anterior pituitary disease such as CD. Nevertheless, the comorbidity of pituitary adenoma and pituicytoma is usually not considered a mere coincidence [4,5]. An unknown pathological transformation or origination from the same lineage is hypothesized, but it lacks concrete evidence.

Although microadenomas were not identified in several cases, most patients achieved remission after surgery, suggesting that pituicytoma might play a role in hyperpituitarism. To date, the explanation that pituicytomas secrete hormones is almost ruled out because they originate from specialized glial cells of neurohypophysis, which cannot secrete anterior pituitary hormones [21].

Two hypotheses have been raised for cases in which ACTH-secreting adenoma could not be histopathologically confirmed. One hypothesis is that an unidentified ACTH-secreting pituitary microadenoma causes hypercortisolism, which also happens in the case of suspected CD without pituicytoma. Studies have reported that approximately 13%–29% of the patients with clinically suspected CD had no adenoma based on histopathological examination, and the patients without confirmed ACTH-secreting adenomas showed a lower remission rate (46% vs. 74%) [18,22]. A common explanation for the missing adenoma is the insufficiency of available specimens for histopathologic examinations. Some surgeons tend to report an excision of adenoma, although it could not be pathologically tested [5]. Another explanation is the existence of a silent hormone-secreting adenoma, such as Crooke cell adenoma. It could cause CD through the secretion of ACTH, but it may be endocrinologically silent [23]. Crooke cell adenomas are typically macroadenomas and behave more aggressively than microsized lesions [21]. In our cases, no Crooke cell was identified. Although most previous cases did not report whether Crooke’s cell was noticed, it is also unlikely that Crooke cell adenomas contributed to CD in these cases with lesions < 10 mm. In brief, a functioning pituitary microadenoma, which might not be included in specimens for final histopathologic diagnosis, is still considered the most likely reason for CD or acromegaly in patients with only pituicytoma.

Another hypothesis regarding the negative finding of hormone-secretory adenoma is that pituicytoma can increase ACTH levels despite the absence of primary adenoma or other lesions. A possible explanation is that PPTs such as pituicytomas might promote the proliferation of adjacent anterior pituitary cells through paracrine signals [3]. In this setting, tumors such as pituicytomas might induce the oncogenic transformation or hyperplasia of hormone-secreting cells. Another explanation is that PPTs might promote anterior pituitary function through the regulation of the hypothalamus, which affects hormonal release [3,14]. As no cytokine or hormone affecting adjacent tissues produced by pituicytomas has been identified, these hypotheses are yet to be studied. It was a pity that CRH staining was not performed in our cases because it is not a routine study in our center, and the small size of our specimens limited us from further staining at present. An immunopathologic study of CRH in pituicytoma specimens from patients with CD might present the possible changes in hypothalamic hormones that might explain the CD without detected corticotroph pituitary adenoma in some cases.

## 5. Conclusions

In conclusion, pituicytoma is a rare low-grade glioma, but it could be suspected even in the presence of significant hyperpituitarism. The misdiagnosis of pituicytoma as pituitary adenoma before surgery is common because pituicytomas can lead to similar symptoms caused by mass effects, such as visual disorders, headache, and anterior pituitary deficiency. Furthermore, pituicytoma might cause hyperfunction such as CD or acromegaly, even without confirmed comorbidity of pituitary adenoma. Thus, although the comorbidity of pituicytoma and suspected CD is rare, a differential diagnosis should be considered. Increasing cases of comorbidity of hyperpituitarism and pituicytoma have been reported in the past 5 years. The comorbidity of pituicytoma and biochemically diagnosed CD has a slightly lower remission rate after the first surgery than does common CD. Intraoperative bleeding might be more common in patients with pituicytoma than in those with CD. Risks of complications such as hypopituitarism are similar in CD and pituicytoma. No recurrence has been noted in the short term, but a long-term prognosis is unexplored.

## Figures and Tables

**Figure 1 jcm-11-04805-f001:**
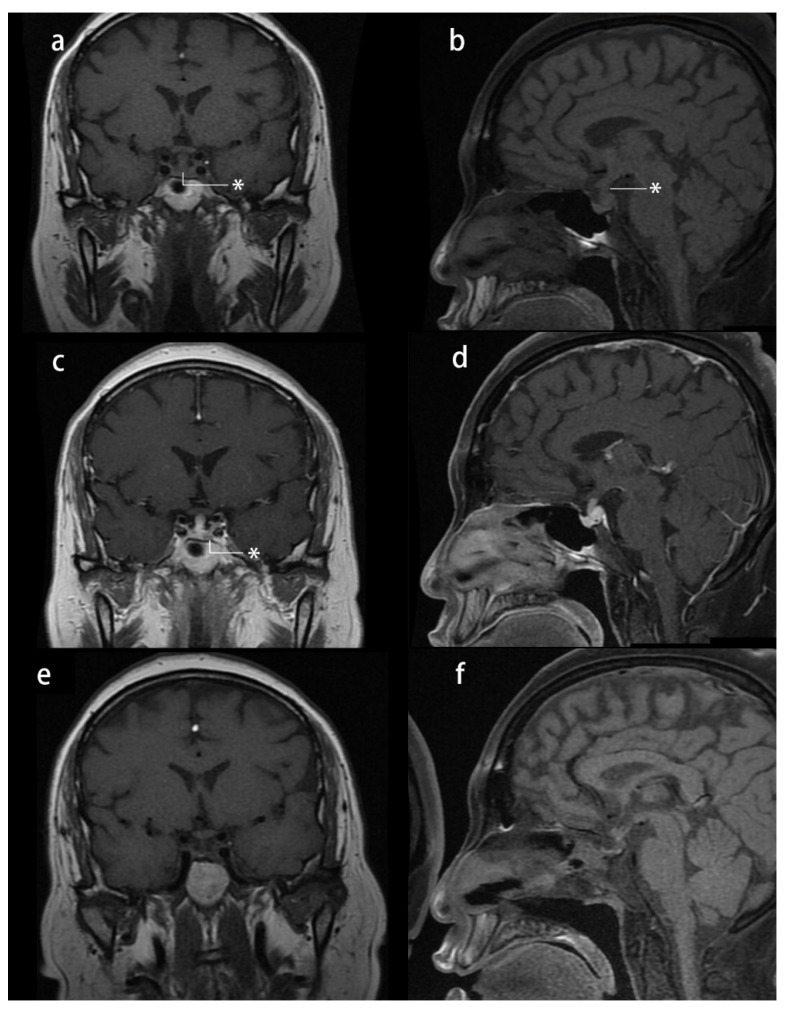
Presurgical and postsurgical pituitary MRI study of case 1: (**a**–**d**) presurgical MRI study: (**a**) T1-weighted image (T1-WI) coronal plane reveals a thickened pituitary stalk *; (**b**) T1-WI sagittal plane reveals a thickened pituitary stalk and loss of hyperintensity * of the T1 signal in the posterior pituitary gland; (**c**) T1-WI contrast-enhanced coronal plane reveals a hypointense nodule * in the left side of the pituitary gland; and (**d**) T1-WI contrast-enhanced sagittal plane reveals a thickened pituitary stalk; (**e**,**f**) one-year postsurgical MRI study: T1-WIs showing a thinner pituitary stalk and loss of hyperintensity of the T1 signal in the posterior pituitary gland.

**Figure 2 jcm-11-04805-f002:**
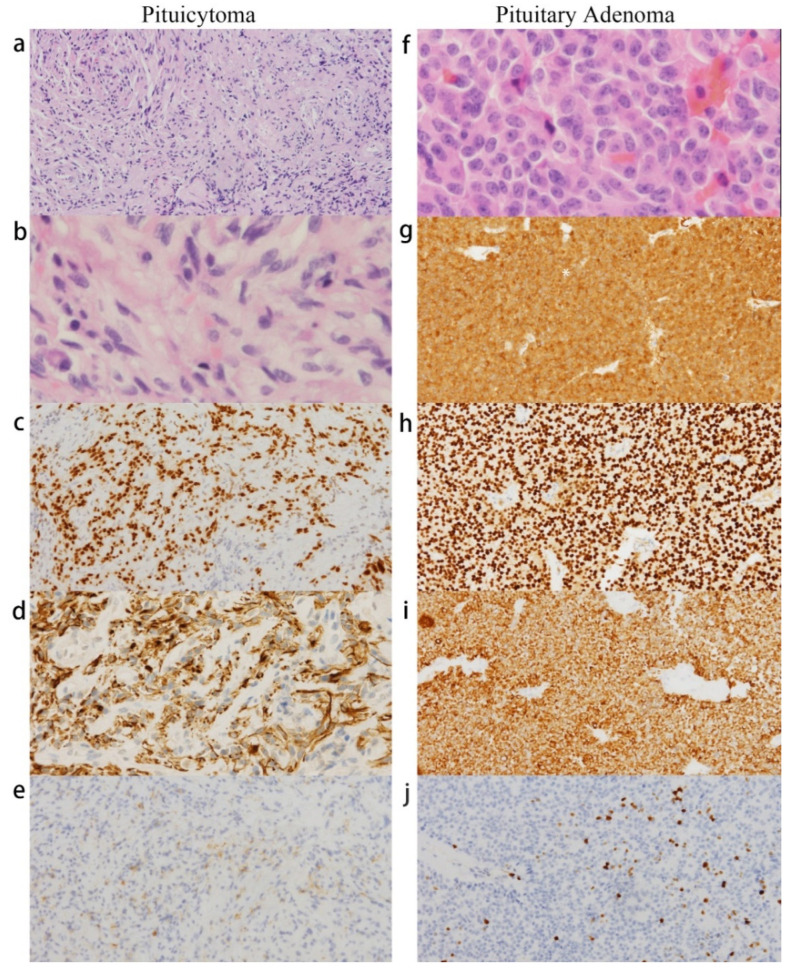
Comorbidity of pituicytoma and pituitary adenoma in case 1: (**a**–**e**) pituicytoma: (**a**,**b**) hematoxylin-and-eosin (H&E) staining shows spindle cells in a mostly fascicular arrangement (**a**, ×100; **b**, ×400); (**c**–**e**) TTF-1, GFAP, and EMA were positive in pituicytoma; (**f**–**j**) pituitary adenoma: (**f**) H&E staining (×400); (**g**,**h**) ACTH and T-PIT were strongly positive; (**i**) AE1/AE3 was positive; and (**j**) increased ki-67 proliferative activity in adenoma (10%).

**Figure 3 jcm-11-04805-f003:**
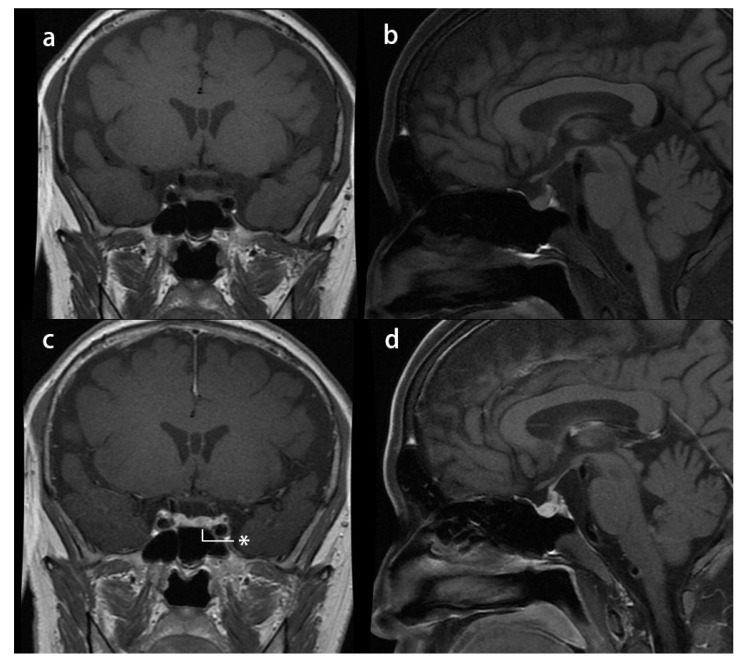
Presurgical pituitary MRI study of case 2: (**a**) T1-weighted image (T1-WI) coronal plane and (**b**) T1-WI sagittal plane showing a normal pituitary stalk without any nodule; (**c**) T1-WI contrast-enhanced coronal plane reveals a hypointense nodule* in the left side of the pituitary gland; and (**d**) T1-WI contrast-enhanced sagittal plane.

**Figure 4 jcm-11-04805-f004:**
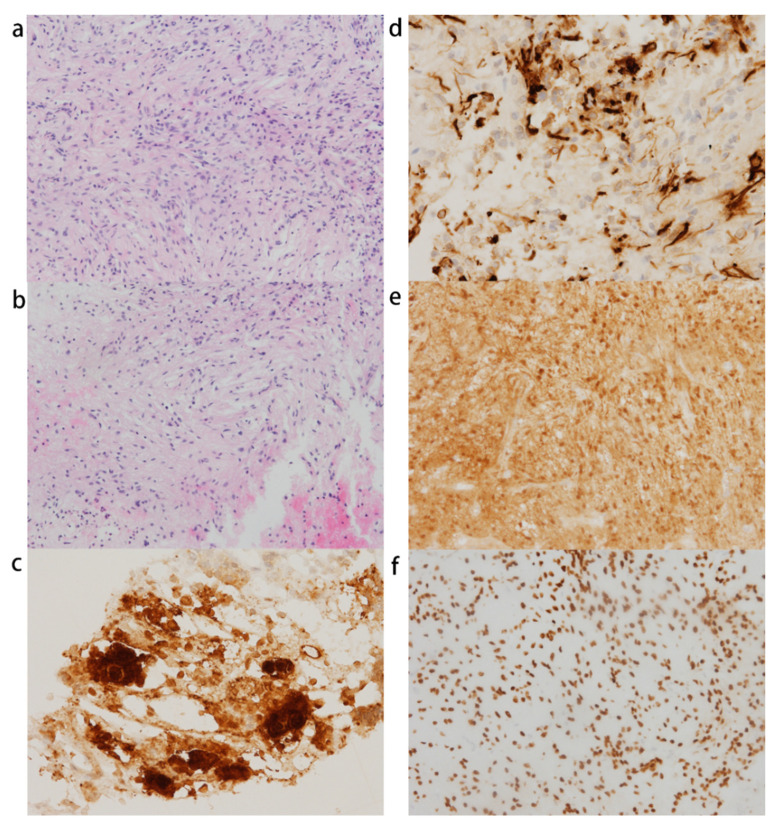
Pituicytoma in case 2: (**a**,**b**) hematoxylin-and-eosin (H&E) staining shows spindle cells in a mostly fascicular arrangement (×200); (**c**) ACTH staining in adjacent pituitary gland cells; (**d**–**f**) GFAP, S-100, and TTF-1 were positive in pituicytoma.

**Figure 5 jcm-11-04805-f005:**
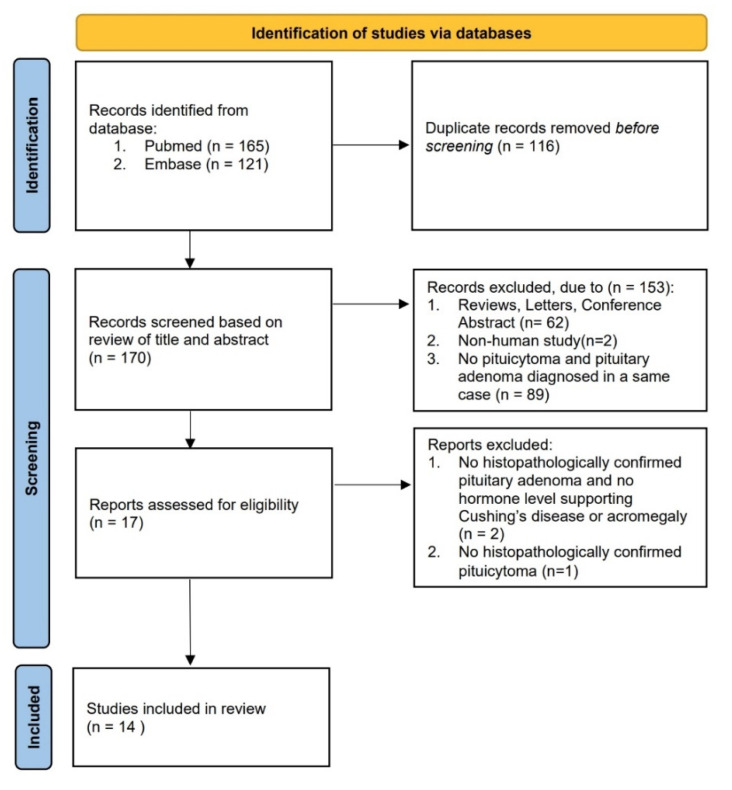
PRISMA flowchart showing the inclusion and exclusion criteria for the review.

**Table 1 jcm-11-04805-t001:** Clinical characteristics of case 1 (51/Female).

Results	Reference	Preoperation	3 Days Postoperation	3 Months Postoperation
Cortisol (µg/mL)	4.0–22.3	30.47	4.9	8.9
ACTH (pg/mL)	0–46	55.7	20.5	12.1
Cortisol: 1mg overnight DST (µg/dL)	<1.8	3.09 (unsuppressed)	/	/
Cortisol: High-dose DST (µg/dL)	Cut-off: 50% on control day	2.09 (suppressed)	/	/
IGF-1 (ng/mL)	87–238	160	189	205
TSH (µIU/mL)	0.38–4.34	0.932	0.961	0.71
FT4 (ng/dL)	0.81–1.98	1.004	1.028	1.09
Testosterone (ng/mL)	0.10–0.75	0.36	<0.1	<0.1
FSH (IU/L)	>40	12.44	12.9	9.15
LH (IU/L)	10.87–58.64	4.02	5.89	1.73
E2 (pg/mL)	<40	28.12	<5	18.42
24 h UFC (µg)	12.3–103.5	186.53	/	/
Serum sodium (mmol/L)	135–145	145	153	143
Serum potassium (mmol/L)	3.5–5.5	3.8	3.2	4.3
Blood glucose (mmol/L)	3.9–6.1	5.3	5.3	3.6
BMI (kg/m^2^)		30.86	/	27.55
Blood pressure (mmHg)		130/80 (with nifedipine)	/	133/91
Lesion size (mm)		9.7 × 4.0, and thickening pituitary stalk	/	/

ACTH: adrenocorticotropin; DST: dexamethasone suppression test; IGF-1: insulin-like growth factor-1; TSH: thyroid-stimulating hormone; FT4: free thyroxine; FSH: follicle-stimulating hormone; LH: luteinizing hormone; E2: estradiol; UFC: urinary-free cortisol; postop: postoperation; BMI: body mass index.

**Table 2 jcm-11-04805-t002:** Clinical characteristics of case 2 (29/Male).

Results	Reference	Preoperation	3 Days Postoperation	3 Months Postoperation
Cortisol (µg/mL)	4.0–22.3	19.7	2.2	1.9
ACTH (pg/mL)	0–46	37.5	<5	9.6
24-h UFC: Low-dose DST (µg)	<12.3	43.7 (unsuppressed)	/	/
24-h UFC: High-dose DST (µg)	Cut-off: 50% on control day	20.8 (suppressed)	/	/
IGF-1 (ng/mL)	117–329	140	/	96
TSH (µIU/mL)	0.38–4.34	1.31	0.203	2.997
FT4 (ng/dL)	0.81–1.98	0.94	1.05	0.92
Testosterone (ng/mL)	1.75–7.81	2.59	1.4	<0.1
FSH (IU/L)	1.27–19.26	6.12	5.73	3.14
LH (IU/L)	1.24–8.62	5.53	4.41	1.15
24 h UFC (µg)	12.3–103.5	185.8	/	<25.8
Serum sodium (mmol/L)	135–145	140	144	145
Serum potassium (mmol/L)	3.5–5.5	3.2	4.1	3.9
Blood glucose (mmol/L)	3.9–6.1	4.3	/	4.4
BMI (kg/m^2^)		22.96	/	21.41
Blood pressure (mmHg)		170/125	/	126/99
Lesion size (mm)		3 × 4	/	/

ACTH: adrenocorticotropin; DST: dexamethasone suppression test; IGF-1: insulin-like growth factor-1; TSH: thyroid-stimulating hormone; FT4: free thyroxine; FSH: follicle-stimulating hormone; LH: luteinizing hormone; E2: estradiol; UFC: urinary-free cortisol; postop: postoperation; BMI: body mass index.

**Table 3 jcm-11-04805-t003:** Summary of patients with suspected Cushing’s disease associated with pituicytomas.

NO.	Publication Year	Author	Age-gender	Tumor Size (mm)	ACTH-Staining Pituitary Adenoma	Pituitary Stalk	Resection	Follow-Up (Month)	Postop. Treatment	Remission	Postsurgical Complications
1	2012	K. Schmalisch [7]	48/M	N/A	(−)	a bulging	N/A	3	reoperation	(+)	hypogonadism
2	2013	S. Chakraborti [12]	24/M	6 × 4	(−)	N/A	GTR	12	N/A	(+)	N/A
3	2015	P. Cambiaso [4]	7/F	N/A	(+)	shortened	STR → GTR	N/A	bilateral adrenalectomy	(−)	N/A
4	2016	X. Guo [9]	46/F	15 × 10 × 7	(−)	thickened	STR	96	radiotherapy	(+)	None
5	2017	V. Barresi [13]	53/F	5 × 6 × 7	(−)	N/A	N/A	16	N/A	(+)	N/A
6	2018	Z. Feng [8]	29/F	4	(−)	N/A	GTR	12	None	(+)	DI, hypoadrenocorticism
7	2018	T.-W. Chang [14]	53/F	5.7 × 5.8 × 4.5	(−)	N/A	N/A	24	N/A	(+)	N/A
8	2018	T.-W. Chang [14]	51/F	6.5 × 6.5 × 7.6	(−)	N/A	N/A	36	radiotherapy	(−)	N/A
9	2018	T.-W. Chang [14]	57/F	5.1 × 2.2 × 3.3	(+)	N/A	N/A	24	N/A	(+)	N/A
10	2018	E. Lefevre [15]	56/F	not visible	(+)	N/A	GTR	3	N/A	(+)	None
11	2019	E. Gezer [5]	37/M	6 × 6.5	(−)	thickened	GTR	N/A	None	(+)	hypopituitarism
12	2019	X. Li [10]	32/F	7.6 × 5.7	(−)	N/A	N/A	49	None	(+)	None
13	2020	F. Marco Del Pont [6]	33/F	N/A	(−)	a swell	GTR	6	None	(+)	N/A
14	2020	A. S. A. L. Rumeh [11]	47/F	5	(−)	N/A	N/A	N/A	None	N/A	N/A

GTR: gross total resection; STR: subtotal resection; DI: diabetes insipidus; N/A: not applicable or not available in original reports.

## Data Availability

Data sharing does not apply to this article because no datasets were generated or analyzed during the current study.

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
