# Peer review of "Pituicytoma Associated with Suspected Cushing’s Disease: Two Case Reports and a Literature Review"

_jcm, 2022, doi:10.3390/jcm11164805_

Round 1

Reviewer 1 Report

Dear authors!

As you had two cases in your institution it seems convincing to publish them together and have a thorough look at this "conincidence". Despite the facts you mentioned it has to be kept in mind that a meaningful pathophysiological connection might not exist and this has to be stated.

Presentation of cases:

Table 1 mixes the data of both cases. Better one table each. The suppression tests you mention in the presentation do not come with your normal values. These data would perfectly fit to such individual tables.

Histology: Did you find Crooke cells in the pituitary specimen? If available CRH staining of the pituicytoma might be helpful for further explanation.

Case#1

Afterwards it became clear that the adrenalectomy was realised because of diagnostic error. This fact has to be pointed at. But do not miss do provide us with the images of histopathology (including the adenoma morphologically suspected). Was the hyperplasia related to the ACTH stimulation - or is an (additional) genetic background possible?

Case#2

Why did you use the desmopressin activation test at sinus petrosus sampling and not a CRH test? What is - retrospectively after operation - your interpretation of the findings of this test?

Table 3: layout is a mess - please correct 

Discussion: In part you relate coincidental descriptions of circumstances mentioned in the few case reports published. This part might be shortened due to a criteria of relevance. Considering "possible mechanisms" you might also integrate information provided by your two cases, considering suggestions mentioned above by my part. Consider also the possible role of hypothalamic hormones in this condition.

Language: Please have an additional check for English language.

Minor: 3.1 p 190: "on average" = median or mean?      5. p 379:  "pituicytoma is a rare benign glioma" do you really mean it is a glioma?

Author Response

We sincerely thank you for thoroughly checking our manuscript and providing very helpful comments to guide our revision. We have addressed each point below:

1. As you had two cases in your institution it seems convincing to publish them together and have a thorough look at this "coincidence". Despite the facts you mentioned it has to be kept in mind that a meaningful pathophysiological connection might not exist and this has to be stated.

        Response: We gratefully thank you for the precious time you spent making constructive remarks! Thank you for pointing out this problem in the manuscript. We added the statement that pituicytoma and pituitary adenoma could develop in one patient without an underlying meaningful pathophysiological connection at the beginning of the discussion section about the possible mechanisms. Since pituitary adenomas are not rare, surgery in the pituitary region might lead to occasional findings of pituicytoma because of symptoms caused by anterior pituitary disease like CD. (Page 13, Line 333-336)

Presentation of cases:

2. Table 1 mixes the data of both cases. Better one table each. The suppression tests you mention in the presentation do not come with your normal values. These data would perfectly fit to such individual tables.

        Response: Thank you for pointing out this problem in our manuscript. We have remade Table 1 as two separate tables (Page 3-4, new Table 1 for case#1 and Table 2 for case#2). The results and references of the dexamethasone suppression test in our center (for 1mg overnight and low-dose DST, serum cortisol 1.8 µg/dL and 24-h UFC 12.3 µg; for high-dose DST, 50% suppression of serum cortisol or 24-h UFC compared with the control day) were added to Table 1 and Table 2.

3. Histology: Did you find Crooke cells in the pituitary specimen? If available CRH staining of the pituicytoma might be helpful for further explanation.

        Response: We appreciate your insightful question about the histopathologic study in our cases. No Crooke cell was suspected in pituitary specimens with routine H&E staining in these two cases.

       Regards the CRH staining of the pituicytoma, it is a pity that CRH was not included as a clinical practice routine in our center, so no CRH staining has been performed on their specimens. Since both of their lesions were small in size, no retained specimens could be used for further CRH staining now. We recognize this limitation and find this idea really interesting, so we added the following statement “It was a pity that CRH staining was not performed in our cases because it is not the routine study in our center, and the small-in-size specimens limited us from further staining at present.” to the discussion (Page 14, Line 381-384), serving as a possible direction for future study. Thank you again for your constructive suggestion!

Case#1

4. Afterwards it became clear that the adrenalectomy was realised because of diagnostic error. This fact has to be pointed at. But do not miss do provide us with the images of histopathology (including the adenoma morphologically suspected). Was the hyperplasia related to the ACTH stimulation - or is an (additional) genetic background possible?

Response: Thank you for your helpful advice! The statement was added to the case presentation section: In case 1, it is clear that she wrongly received an adrenalectomy at first because of a diagnostic error, mistaking the nodules in the adrenal gland as the responsible lesion. (section 4.3, Line 311-313). However, it is a pity that she received the adrenalectomy in another local center before being further referred to our department. Therefore, we had no access to the original histopathology image of her adrenal lesion. Thus, it was also hard to determine whether the hyperplasia of the adrenal gland was primary or was only related to the ACTH hypersecretion in Case #1.  

Case#2

5. Why did you use the desmopressin activation test at sinus petrosus sampling and not a CRH test? What is - retrospectively after operation - your interpretation of the findings of this test?

Response: Thank you for pointing out this issue. We did not perform a CRH test because CRH for the clinical test is not currently available in public hospitals in China by far (partially for cost reasons). In this situation, a Chinese consensus about Cushing’s syndrome (DOI: 10.3760/cma.j.issn.0376-2491.2016.11.002, original text in Chinese, published in 2015) suggested that desmopressin could serve as the substitute for CRH in BIPSS. In clinical practice in our center (The Optimal Cut-off of BIPSS in Differential Diagnosis of ACTH-dependent Cushing’s Syndrome: Is Stimulation Necessary? J Clin Endocrinol Metab. 2020, DOI: 10.1210/clinem/dgz194), we observed a similar sensitivity and specificity of IPSS with the desmopressin activation test in the diagnosis of CD.

        The routine immunopathologic study failed in confirming corticotroph hyperplasia or adenoma in Case #2. To better present the pathologic result in case #2, a new Figure 4 (Page 8) showing the immunopathologic image of case #2 was also added. Retrospectively after the operation, we still considered the diagnosis of this patient to be CD, and the most possible reason might be ACTH-secreting adenoma. First, the central-to-peripheral ACTH gradient was 55 after the activation of desmopressin. Considering the specificity of this test is almost 100% when using 3 as the cut-off, it strongly supported the diagnosis of CD. Second, he did achieve remission in Cushing’s syndrome after surgery, and no recurrence has been observed by far. The insufficiency of available specimens for histopathologic examinations might explain the “missing microadenoma”, as the overall lesion size was < 5mm and it was also susceptible to damage during the surgery.

Similarly, only 3 out of 14 previous cases reported a confirmed ACTH-secreting adenoma detected during the pathological examination. Nevertheless, although many researchers considered the undetected corticotroph adenoma as the most likely cause of CD, other explanations remain (such as an underlying role of pituicytoma). It is still a matter of debate, and we discussed it in section 4.4 (Page 14, Paragraph started at Line 348 about the underlying “missing microadenoma”).

6. Table 2: layout is a mess - please correct

Response: Thank you for pointing out this problem in our manuscript, we feel sorry for the inconvenience in reading. Table 2 was remade as New Table 3 (since we spilt original table 1 into table 1 and table 2). Considering the suggestion given by another reviewer, we removed the cases related to acromegaly or nonfunctioning tumors (Case #5, #12, #13, and #18 in the original Table 2) from the table. In the modified Table 3, only 14 cases of pituicytoma associated with CD were presented. Other 3 Cases with acromegaly and 1 Case with nonfunctioning adenoma were briefly described in the manuscript. The main modifications are as follows:

  • Accordingly, the title of Table 3 was edited to Summary of patients with Cushing’s disease associated with pituicytomas.
  • In table 3, The columns presenting “pre-operation diagnosis” and “tumor location” were deleted. The column name “Confirmed pituitary adenoma” was changed to “ACTH-staining pituitary adenoma”, and “Outcomes” was changed to “Remission”. The descriptions in columns “Remission” and “postsurgical complications” were simplified.

        We hope this revised table would be easier to read.

7. Discussion: In part you relate coincidental descriptions of circumstances mentioned in the few case reports published. This part might be shortened due to a criteria of relevance. Considering "possible mechanisms" you might also integrate information provided by your two cases, considering suggestions mentioned above by my part. Consider also the possible role of hypothalamic hormones in this condition.

Response: Thank you for the constructive suggestions. We have made the changes as follows:

  • At the beginning of the “possible mechanisms” part, we strengthened that there is the possibility that pituicytoma and pituitary adenoma might have no underlying pathophysiological connection (Section 4.4) as you suggested above.
  • We shorten the part relevant to other posterior pituitary tumors (like granular cell tumors).
  • We revised our cases in this section (Line 361-362: no identified Crooke cell; Line 383-384, lack of CRH staining in our specimens).
  • Regards the possible role of hypothalamic hormones, although there are relative hypotheses, no concrete research associated with pituicytoma and hormones (including CRH and ACTH) has been performed by far to our knowledge. We added this discussion for future study: “Immunopathologic study of CRH in pituicytoma specimens from patients with CD might present the possible change of hypothalamic hormones, which might explain the CD without detected corticotroph pituitary adenoma.” (Page 14, Line 384-386).

8. Language: Please have an additional check for English language.

Minor: 3.1 p 190: "on average" = median or mean?      5. p 379:  "pituicytoma is a rare benign glioma" do you really mean it is a glioma?

Response: Thank you so much for your careful check. All mentioned ambiguities have been corrected as follows, as well as an additional check for the English language.

        In 3.1 p 190: "on average" (follow-up period: 18 months) was corrected as “the mean follow-up period”.

     In 5. p 379: "pituicytoma is a rare benign glioma", we did intend to mean that pituicytoma is rare glioma with a relatively good prognosis among all glioma. The word “benign” was not exact enough, and the description has been corrected as “pituicytoma is a rare low-grade glioma”.

      We sincerely hope that this revised manuscript has addressed all your constructive comments and suggestions. Thank you again for taking the time to review our manuscript so carefully!

Reviewer 2 Report

The present article describes the clinical course of two patients with pituicytoma and Cushing's disease; a 51-year-old female with histologically proven ACTH-secreting pituitary microadenoma; and another 29-year-old male patient without histologic demonstration of the tumor. In addition, the authors review the literature regarding the association of these two entities.

Comments

  1. In the title, change the term "comorbid with" to "associated with". Also, do this throughout the manuscript.

  1. Was the possibility of neuroinfundibulitis as a cause of diabetes insipidus ruled out in patient #1, and were any studies specifically directed in this direction?.

  1. The result of the Nugent's test in patient #2 should be provided.

  1.  It would be convenient to provide a figure with the histopathological study of the pituicytoma in patient #2.

  1. In a recent article published by Iglesias et al. (Endocrine (2020) 70:15-23), 15 patients with Cushing's disease and 6 with acromegaly associated with pituicytoma were described, i.e. a total of 21 patients; however, in this most recent literature review only 18 patients with pituicytoma are reported, 14 of them with Cushing's disease and 3 with acromegaly. Please comment.

  1. This case should be considered: Case Reports Endocrinol Diabetes Nutr (Engl Ed) . 2020 Nov 22;S2530-0164(20)30205-6. doi: 10.1016/j.endinu.2020.06.005.

  1. Table 2 is difficult to interpret as it currently appears. Please improve its presentation to make it easier to understand and read.

  1. Because the authors present two patients with pituicytoma and Cushing's disease and review the literature on this association, it does not make much sense in the article to consider cases of pituicytoma associated with somatotropinomas or nonfunctioning pituitary adenomas. Please remove this information throughout the manuscript and in Table 2.

  1. With concerning the Discussion section, it would seem more appropriate to first consider the prevalence of the association of pituicytoma with Cushing's disease, followed by clinical, diagnostic, and therapeutic aspects.

  1. Finally, the Limitations section does not seem very appropriate in this case, since the manuscript refers to a clinical description of patients and a review of the literature.

Author Response

Thank you for providing feedback on our manuscript and we are grateful for the insightful comments on and valuable improvements to our paper. Below we detail the changes made in our revision according to your comments.

  1. In the title, change the term "comorbid with" to "associated with". Also, do this throughout the manuscript.

Response: According to your valuable advice, the title has been changed to “Pituicytoma associated with Cushing’s disease: two case reports and literature review”, together with another 7 changes from “comorbid with” to “associated with” in the manuscript.

  1. Was the possibility of neuroinfundibulitis as a cause of diabetes insipidus ruled out in patient #1, and were any studies specifically directed in this direction?

Response: Thank you for your rigorous comment. We totally understand your concern that diabetes insipidus in patient #1 was most likely the consequence of the mass effect or stalk effect caused by pituicytoma or pituitary adenoma. In a previous review of pituicytoma (Clinical features, diagnosis and therapy of pituicytoma: an update, 2019, DOI: 10.1007/s40618-018-0923-z), about 5% of patients were comorbid with DI before surgery. However, as mentioned in the literature review section of the manuscript, diabetes insipidus has not been reported in any of the patients with pituicytoma associated with pituitary adenoma by far. Studies on DI in pituicytoma, especially in pituicytoma associated with other pituitary adenomas, are still limited.

  1. The result of the Nugent's test in patient #2 should be provided.

   Response: Thank you so much for your careful check. According to your suggestion, the result of Nugent’s test of patient #2 was added to the manuscript (Page 6, Line 135). We did not include this result in our original manuscript because he only received an overnight 1mg dexamethasone suppression test in another center before being referred to our department, which indicated Cushing’s Syndrome: the overnight 1mg dexamethasone suppression test could not suppress serum cortisol (16.6 µg/dL). Then, in our department, we did not repeat the Nugent’s test but performed a low-dose and a high-dose DST to further confirm the diagnosis of CD. Therefore, the available result of Nugent’s test in patient #2 was from another center.

  1. It would be convenient to provide a figure with the histopathological study of the pituicytoma in patient #2.

Response: We gratefully appreciate your valuable suggestion. A new Figure 4 presenting the histopathological image of pituicytoma in patient #2 has been inserted on Page 8. As no confirmed ACTH pituitary adenoma was observed in the histopathological study, we only presented the section of pituicytoma in Figure 4, showing the image of H&E, ACTH, GFAP, S-100, and TTF-1 staining.

  1. In a recent article published by Iglesias et al. (Endocrine (2020) 70:15-23), 15 patients with Cushing's disease and 6 with acromegaly associated with pituicytoma were described, i.e. a total of 21 patients; however, in this most recent literature review only 18 patients with pituicytoma are reported, 14 of them with Cushing's disease and 3 with acromegaly. Please comment.

Response: Thank you for reading our paper carefully and raising this question. In the mentioned article (Iglesias et al., Adenohypophyseal hyperfunction syndromes and posterior pituitary tumors: prevalence, clinical characteristics, and pathophysiological mechanisms, Endocrine (2020) 70:15-23), 15 patients with CD and 6 with acromegaly were reviewed. However, these cases were associated with overall posterior pituitary tumors (PPT), which included not only pituicytoma but also granular cell tumor (GCT), spindle cell oncocytoma (SCO), and sellar ependymomas (SE). (In 2017, the WHO established posterior pituitary tumors as an entity class, including pituicytoma, GCT, SCO, and SE). Since our literature review only concentrated on cases associated with pituicytoma, the cases included were fewer than in this previous literature review.

  1. This case should be considered: Case Reports Endocrinol Diabetes Nutr (Engl Ed). 2020 Nov 22;S2530-0164(20)30205-6. DOI: 10.1016/j.endinu.2020.06.005.

Response: Thank you for your recommendation. We read this case report (López-Muñoz, Beatriz et al. Concurrent corticotroph pituitary tumor and granular cell tumor: A very uncommon association. Endocrinologia, diabetes y nutricion, 2020, doi:10.1016/j.endinu.2020.06.005) and noticed that it presented a rare case of Cushing’s disease associated with granular cell tumor (GCT). As mentioned above in comment #5, GCT is different from pituicytoma in classification (although both these two tumors are rare posterior pituitary tumors located in the posterior pituitary). Because our study focused on pituicytoma, this case was not included in our final literature review. Thank you again for your kind recommendation of this rare case.

  1. Table 2 is difficult to interpret as it currently appears. Please improve its presentation to make it easier to understand and read.

Response: Thank you for pointing out this problem in our manuscript. We have remade Table 2 (as Table 3 in the revision, because Table 1 was also remade as two separate tables) to clearly present cases of Cushing’s disease associated with pituicytoma, and the main modifications are as follows:

  • Considering your valuable advice in comment #8, we removed the cases related to acromegaly or nonfunctioning tumors (Case #5, #12, #13, and #18 in the original Table 2). In the modified Table 3, only 14 cases of pituicytoma associated with CD were presented.
  • Accordingly, the title of Table 3 was edited to Summary of patients with Cushing’s disease associated with pituicytomas.
  • The columns presenting “pre-operation diagnosis” and “tumor location” were deleted. The column name “Confirmed pituitary adenoma” was changed to “ACTH-staining pituitary adenoma”, and “Outcomes” was changed to “Remission”. The descriptions in columns “Remission” and “postsurgical complications” were simplified.

We hope it would be easier to read after improvement.

  1. Because the authors present two patients with pituicytoma and Cushing's disease and review the literature on this association, it does not make much sense in the article to consider cases of pituicytoma associated with somatotropinomas or nonfunctioning pituitary adenomas. Please remove this information throughout the manuscript and in Table 2.

Response: Thank you for the above suggestion. We agreed that this review should concentrate more on pituicytoma associated with CD. Therefore, cases associated with acromegaly or nonfunctioning pituitary adenomas have been removed from Table 3 (Case #5, #12, #13, and #18 in the original Table 2).

We also removed most information related to somatotropinomas and nonfunctioning pituitary adenomas in the manuscript, including the clinical features of 4 cases in detail, and the discussion related to their features. We only retained a brief description of other pituitary adenomas in Review of the literature section (Page 8, Line 190-191), discussion sections 4.1(Page 11, Line 246-248), and 4.2 (Page 12, Line 267-269). It described the fact that only 3 acromegaly cases and 1 nonfunctioning case were reported in the previous study, and the tumors were larger than those in CD in size. It is possible that cases of pituicytomas associated with ACTH-secreting pituitary adenomas are much easier to be noticed because of Cushingoid symptoms, while pituicytomas associated with somatotropinomas and nonfunctioning pituitary adenomas were difficult to observe when the lesion is relatively small. We hope it might help if readers wonder whether CD is the most common hyperpituitarism associated with pituicytoma. Thank you again for your kind advice.

  1. With concerning the Discussion section, it would seem more appropriate to first consider the prevalence of the association of pituicytoma with Cushing’s disease, followed by clinical, diagnostic, and therapeutic aspects.

Response: Thank you for your helpful suggestion. We have improved the structure and organization of the Review of Literature and the Discussion section. The main modifications are as follows:

  • In the Review of Literature section 3.1 (Page 11), we presented the prevalence first, then followed by clinical symptoms, radiological findings, treatments, immunopathologic studies, and outcomes after treatment.
  • the Discussion section was reorganized in order: 1. Prevalence and Symptoms: Pituicytoma and Pituicytoma associated with Cushing’s Disease; 4.2. Radiologic and Immunopathologic feature; 4.3. Treatment and Prognosis: Compared with Cases with Typical Pituitary Adenomas; 4.4. Pituicytoma and CD: Possible Mechanisms.

    In addition to these, some modifications in words were made. We hope it was much easier for reading. Thank you for your helpful suggestion.

  1. Finally, the Limitations section does not seem very appropriate in this case, since the manuscript refers to a clinical description of patients and a review of the literature.

Response: Thank you for your constructive suggestion. The previous Limitations section was reorganized into the Discussion section, which was described as the limited knowledge about CD associated with pituicytomas in available studies. (Page 14, Line 387-401)

         We sincerely hope that this revised manuscript has addressed all your constructive comments and suggestions. Thank you again for taking the time to review our manuscript so carefully!

Reviewer 3 Report

Previous cases reporting , like this one, associations between pituicytoma and CD have been published. These cases are rare and worth being reported.

However, several aspects are really confusing:

  1. The association described as not being reported before between pituicytoma, DI and CD is exaggerated as DI is most likely the consequence of the presence of pituicytoma and not a third disease on its own
  2. The case in whom apparent postsurgical cure of CD was obtained but no ACTH staining was described is puzzling and an explanation must exist and must be offered to the reader

Author Response

We sincerely thank you for providing very helpful comments to guide our revision, which have helped improve our manuscript. We have addressed each point below:

  1. The association described as not being reported before between pituicytoma, DI and CD is exaggerated as DI is most likely the consequence of the presence of pituicytoma and not a third disease on its own.

Response: We gratefully appreciate your valuable suggestion. We totally understand your concern that DI is most likely the consequence of the mass effect or stalk effect caused by pituicytoma. In a previous review of overall pituicytoma (Clinical features, diagnosis and therapy of pituicytoma: an update, 2019, DOI: 10.1007/s40618-018-0923-z), about 5% of patients were comorbid with DI before surgery (also described in Line 244-246 in our manuscript). In 18 cases related to pituitary adenoma, no DI before surgery has been reported yet. The description of case #1 as the first case reporting pituicytoma, DI, and CD in the abstract was deleted to avoid potential exaggerated expression.

  1. The case in whom apparent postsurgical cure of CD was obtained but no ACTH staining was described is puzzling and an explanation must exist and must be offered to the reader.

Response: Thank you for pointing out this issue in our manuscript. In case #2, the immunopathologic study did fail in confirming the immunopathologic diagnosis of ACTH staining-positive pituitary adenoma. However, in clinical practice, this phenomenon is not rare. According to several previous studies (Nishioka, H. and S. Yamada, Cushing's Disease. Journal of clinical medicine, 2019. 8(11); Carr, S.B., et al., Negative surgical exploration in patients with Cushing's disease: benefit of two-thirds gland resection on remission rate and a review of the literature. Journal of neurosurgery, 2018. 129(5) ), approximately 13%–29% of the patients with confirmed CD had no adenoma based on histopathological examination, but around 46% of these CD patients still achieved remission after surgery. As a comparison, in 14 patients with pituicytoma associated with Cushing’s disease (reviewed in Table 2), only 3 of them confirmed the ACTH staining-positive pituitary adenoma in the immunopathologic study. The positive rate of ACTH-secreting pituitary adenoma in patients with CD associated with pituicytoma was much lower than in common CD patients.

         In brief, a negative finding of pituitary adenoma in CD based on histopathological examination does not mean denying the clinical diagnosis of ACTH adenoma. We discussed the possible reasons for the negative staining of ACTH pituitary adenoma in patients with CD in section 4.4. of the manuscript (Pituicytoma and CD: Possible Mechanisms, Page 14, Line 341-369).

    One hypothesis is that the ACTH-secreting adenoma did exist but the histopathological result failed in observing them. A common explanation for the ‘missing adenoma’ is the insufficiency of available specimens for histopathologic examinations. Another explanation is the existence of a silent hormone-secreting adenoma like Crooke cell adenoma, which could cause CD but might be negative in staining of ACTH in some statuses. Another hypothesis is that pituicytoma could somehow increase ACTH levels, despite the absence of primary adenoma or other lesions in the sellar region. Some researchers suggested that pituicytoma might promote the proliferation of adjacent anterior pituitary cells through some paracrine signals (Iglesias, P., et al., Adenohypophyseal hyperfunction syndromes and posterior pituitary tumors: prevalence, clinical characteristics, and pathophysiological mechanisms. Endocrine, 2020), or promote anterior pituitary function through the regulation of the hypothalamus (Chang, T.-W., et al., Correlations between clinical hormone change and pathological features of pituicytoma. British journal of neurosurgery, 2018). However, these hypotheses are yet to be studied by far. Future studies might pay more attention to the possible role of hypothalamic hormones. For example, an immunopathologic study of CRH in pituicytoma specimens from patients with CD might show the possible change of hypothalamic hormones, which might explain the CD without detected corticotroph pituitary adenoma in some cases. (Page 14, Line 383-386)

        Thank you again for your helpful comments and suggestions. We hope our revisions could address them all.

Reviewer 4 Report

The authors presented two rare cases of pituicytoma coincidence with Cushing’s disease together with extensive literature review. The layout of the tables is unreadable and it makes data unclear. There are discrepances in the size of the lesion in case 1. Another questions are: why prednisone in replacement therapy? why hypopituitarism following microadenoma neurosrgery?

Author Response

We sincerely than you for thoroughly checking our manuscript and providing very helpful comments to guide our revision. We have addressed each point below:

  1. Modification of layout of Tables

Response: Thank you for pointing out this problem in our manuscript. We have revised Table 1 and Table 2 to clearly present cases of Cushing’s disease associated with pituicytoma, and the main modifications are as follows:

  • Table 1 was split into two tables (New Table 1 and Table 2). Each table presented the clinical features of one case (Table 1 for case 1, and Table 2 for case 2).
  • Table 2 was revised as New Table 3. Considering the suggestion given by another reviewer, we removed the cases related to acromegaly or nonfunctioning tumors (Case #5, #12, #13, and #18 in the original Table 2). In the modified Table 3, only 14 cases of pituicytoma associated with CD were presented. Other 3 Cases with acromegaly and 1 Case with nonfunctioning adenoma were briefly described in the manuscript.
  • Accordingly, the title of Table 3 was edited to Summary of patients with Cushing’s disease associated with pituicytomas.
  • In table 3, The columns presenting “pre-operation diagnosis” and “tumor location” were deleted. The column name “Confirmed pituitary adenoma” was changed to “ACTH-staining pituitary adenoma”, and “Outcomes” was changed to “Remission”. The descriptions in columns “Remission” and “postsurgical complications” were simplified.

We hope the revised tables would be easier to read. Thank you for you suggestion.

  1. Discrepancies in the size of the lesion in case 1

Response: Thank you so much for your careful check. Did the mentioned discrepancies in the size of the lesion in case 1 refer to the differences between the size reported in dynamic MRI (9.7 × 4.0 mm lesion, Page 4, Line 102) before transsphenoidal surgery and the size reported in the pathologic study (9 × 5 × 4 mm resected specimen, Page 4, Line 117)? We have checked all sizes of the lesion in case 1 again, and all of them were consistent with the results in the original radiologic report and specimen size in the pathologic report. This “discrepancy” was highly likely caused by different accuracy and measuring methods performed between MRI and the pathologic study. Please contact us if there are any problems related to the case presentation. Thank you again for your comments!

  1. Why Prednisone in replacement therapy?

Response: Thank you for your question. Although hydrocortisone (or cortisone acetate) is recommended as the first choice for replacement therapy in adrenocortical insufficiency, some guidelines (DOI: 10.1210/jc.2015-1710; DOI:10.1507/endocrj.ej16-0242) also mentioned that long-acting drugs like prednisone could be used as an alternative to hydrocortisone, especially in patients with reduced compliance. Compared to hydrocortisone, some patients show better medication adherence when using long-acting drugs.

In clinical practice in our center, there are also some patients who strongly prefer prednisone for their replacement therapy to hydrocortisone, mostly because of the convenience (reducing the frequency of taking pills). In our manuscript, Case #1 preferred prednisone in her replacement therapy, while Case #2 took hydrocortisone acetate as in most patients with adrenal insufficiency.

  1. why hypopituitarism following microadenoma neurosurgery?

Response: Thank you for pointing out this issue. Both two cases reported in our manuscript did develop hypopituitarism after transsphenoidal surgery and received replacement therapy during the 17 months (case #1) and 8 months (case #2) follow-up. Regards the explanation for hypopituitarism, case #1 and case #2 might have different causes.

         For case #1, central diabetes insipidus existed before surgery, which suggested a pituitary stalk injury in addition to pituicytoma and corticotroph pituitary adenoma. In MRI, a thickened pituitary stalk also indicated the lesion of the pituitary stalk. With this lesion, the patient in case #1 was at a higher risk for abnormal pituitary function. Additionally, since the cause of DI was unclear, a biopsy in the posterior pituitary was also performed besides resecting the mass of the left side of the pituitary gland. The biopsy might also lead to extra pituitary function damage when compared with common microadenoma neurosurgery.

         For case #2, no hypopituitarism was identified before surgery. Thus, transsphenoidal surgery was likely the only explanation for hypopituitarism after surgery, although the lesion was relatively small in size. We referred to 14 previously reported cases with CD associated with pituicytoma and noticed that only 6 cases described complications after surgery in detail. 3 out of 6 developed hypopituitarism and received a relatively long-term replacement therapy, although all of them also had a pituitary lesion < 10mm and underwent microadenoma neurosurgery. Thus, it is possible that CD associated with pituicytoma is at a higher risk for hypopituitarism when compared with common corticotroph pituitary adenoma because of unclear features of pituicytoma. However, in the light of a limited number of reported cases by far, we could not conclude whether hypopituitarism in case #2 was merely an uncommon complication following microadenoma surgery. Additionally, the patient of case #2 was followed up for 8 months at the latest visit, which was a relatively short period. We will observe whether the hypopituitarism could be improved in a long-term follow-up.

We sincerely hope that this revised manuscript has addressed all your constructive comments and suggestions. Thank you again for taking the time to review our manuscript!

Round 2

Reviewer 3 Report

The previous suggestions were adequately adressed. However, the discussion section is now way too long and confusing- in my opinion it needs to be changed to a significantly shorter and more concise version.

Author Response

Response: Thank you for providing feedback on our revision, and we are grateful for your comments on our manuscript. According to your suggestion, we revised the discussion section to a shorter version (within 3 pages). Below we detail the main modifications made in our revision:

  1. In section 4.1 (prevalence and symptoms), we revised the description of pituicytoma to a shortened version, and deleted sentences about the concomitant pituicytoma and pituitary microadenoma (Line 259-261 in the previous version).
  2. In section 4.2 (Radiologic and Immunopathologic feature), we concisely described the mean size of lesions of pituicytoma associated with CD (Line 261) instead of the original detailed description of the size of pituicytoma in all cases.
  3. In section 4.3 (Treatment and Prognosis): 
    • Regarding the remission rate after surgery, we reorganized it at the beginning of section 4.3 and shorten it by 4 lines;
    • We deleted a sentence describing a previously reported case about the easily bleeding lesion during surgery;
    • We revised the part about anterior hypopituitarism after surgery and shortened it by 3 lines.
    • In the part discussing the possible explanations for a thinner pituitary stalk of case #1, we deleted a sentence describing the absence of previous reports.
  4. In section 4.4 (Pituicytoma and CD: possible mechanisms):
    • We removed a sentence with similar meanings to others (highlighting that pituicytoma could not secrete anterior pituitary hormones by itself);
    • We removed the detailed description of Crooke cell adenoma and shortened the relative discussion since it was unlikely the underlying cause for ‘missing adenoma’ in cases with lesions < 10 mm;
    • We deleted the last paragraph about the limitations of previous studies, in which most content has been presented in earlier sections.

        In addition, the expression of the discussion part has also been improved to make the sentences more concise. We sincerely hope this revised discussion section could be clearer for reading. Thank you again for taking the time to review our manuscript so carefully!